# A Successful New Case of Twin Pregnancy in a Patient with Swyer Syndrome—An Up-to-Date Review on the Incidence and Outcome of Twin/Multiple Gestations in the Pure 46,XY Gonadal Dysgenesis

**DOI:** 10.3390/ijerph19095027

**Published:** 2022-04-20

**Authors:** Izabela Winkler, Ilona Jaszczuk, Marek Gogacz, Piotr Szkodziak, Tomasz Paszkowski, Katarzyna Skorupska, Michał Ciebiera, Maciej Skrzypczak

**Affiliations:** 1Second Department of Gynecological Oncology, St. John’s Center of Oncology of the Lublin Region, 7 Jaczewskiego Street, 20-090 Lublin, Poland; 2Second Department of Gynecology, Medical University of Lublin, 20-954 Lublin, Poland; gogacz@yahoo.com (M.G.); kasiaperzylo@hotmail.com (K.S.); skrzypczakmk8@gmail.com (M.S.); 3Department of Pediatric Hematology, Oncology and Transplantology, Children Clinical Hospital, II Department of Pediatrics, Medical University of Lublin, 20-093 Lublin, Poland; ilona.krupa@interia.pl; 4Third Department of Gynecology, Medical University of Lublin, 20-954 Lublin, Poland; piotr.szkodziak@gmail.com (P.S.); tp101256@gmail.com (T.P.); 5Second Department of Obstetrics and Gynecology, Center of Postgraduate Medical Education, 01-809 Warsaw, Poland; michal.ciebiera@gmail.com

**Keywords:** Swyer syndrome, twin pregnancy, pure gonadal dysgenesis

## Abstract

Background: The aim of the present study is to report a rare occurrence of a successful twin pregnancy in a woman with pure 46,XY gonadal dysgenesis. Result(s): A patient with Swyer syndrome (pure 46,XY gonadal dysgenesis) presented with a twin pregnancy after in vitro fertilization. Due to unidentified conditions, the patient developed selective intrauterine growth restriction in one of the fetuses. Twins were born at 33 weeks of pregnancy due to the risk of asphyxia. Nonetheless, the patient did not develop gonadal malignancies before the pregnancy and, despite receiving estrogen, remained amenorrheic. Conclusion(s): The aim of this case report is to show the course of twin pregnancy in patients with Swyer syndrome through assisted reproduction. Due to certain disorders in the development of their reproductive organs, such as the less mature uterus, such pregnancies may be associated with an increased risk. The above case report demonstrates the need to systematize methods of pregnancy management in patients with Swyer syndrome, such as: preparation for the pregnancy, assessment of the uterus, medications used, and necessary checkups. Capsule: This case report and review shows clinicians that patients with Swyer syndrome may become pregnant. Twin pregnancies may occur without any major problems through assisted reproduction.

## 1. Introduction

Swyer syndrome, also known as 46,XY pure gonadal dysgenesis, is a rare condition that affects sexual development. It was described by Swyer in 1955 [1]. It is classified as a disorder of sex development (DSD), or a situation in which chromosomal, gonadal or anatomic sex development is abnormal. It occurs in approximately 1 in 80,000 people depending on the described population. Patients with Swyer syndrome present with typical female external genitalia. The uterus and fallopian tubes of such patients may be smaller or normally formed, but the gonads are not functional. Instead of sex glands, women with Swyer syndrome present with gonadal streaks, i.e., the ovaries are replaced by a form of fibrous tissue [2]. Between 10% and 20% of women with Swyer syndrome have a deletion in the DNA-binding region of the *SRY* gene, while in the remaining 80–90% of cases, the *SRY* gene is normal, and mutations in other testis-determining factors (*MAP3K1*, *DHH*, *NR5A1*) are present [3]. 

## 2. A Case Report

In this manuscript, we want to present the case of a dichorionic diamniotic twin pregnancy that occurred in a patient with Swyer syndrome who was treated with assisted reproductive techniques. 

The described patient was diagnosed with Swyer syndrome at 15 years of age due to primary amenorrhea [4]. Following diagnosis by karyotype, she underwent removal of the streak gonads to minimalize the risk of malignancy in the future. The patient presented with a female phenotype, external genitalia, and female hair on her body and in the genital area. During the periconception period, the patient’s body mass index (BMI) was 29.37 kg/m^2^. She was 177 cm tall and weighed 97 kg. Her breast development was also normal. A pelvic ultrasound examination revealed a uterus with a longitudinal diameter of 60 mm, anteroposterior diameter of 35 mm, and transverse diameter of 39 mm. The patient had been using hormone replacement therapy (HRT) since the age of 18 and had regular cycles. Most of the time, until the age of 38, she had been using 2 mg of estradiol with 0.5 mg of norgestrel. The patient also presented with hypothyroidism, and she took oral levothyroxine 50 mcg daily. A detailed description of the drugs used during pregnancy can be found in Table 1.

Moreover, the hypoplastic uterus in patients with Swyer syndrome may be able to sustain pregnancy. One year before starting the donor egg procedure, the abovementioned HRT was changed to 1.5 mg of estradiol with 2.5 mg of nomegestrol. The patient underwent an office hysteroscopy in order to assess the shape and condition of the endometrial cavity (Figure 1). The histopathological samples at the time of the hysteroscopy were characterized by an atrophic endometrium. After the hysteroscopy, the patient was referred to the ART center, where the egg donor procedure was performed. 

The patient received 150 mg of acetylsalicylic acid daily to decrease the risk of pre-eclampsia. Her blood pressure was normal (below 140/90 mmHg) throughout the pregnancy. Moreover, the patient used estrogens and progesterone intravaginally and orally (we have provided the exact hormonal treatment scheme during each trimester in Table 1). We checked her liver parameters each month due to several risk factors associated with pre-eclampsia. Liver enzymes (AST aspartate aminotransferase and ALT alanine aminotransferase) were elevated to approximately 60 UL/l at weeks 26/27 of pregnancy. Therefore, we decreased the dose of hormones. In addition, the patient used prednisone 5 mg orally per day. 

During the first and second trimesters, the twins developed normally. Around 28 weeks of pregnancy, the discrepancy between the estimated fetal weight equaled 18.6%, but during weeks 31/32, it rose to 27.1%. At 32 weeks and 5 days, the dichorionic diamniotic (Figure 2) twin pregnancy was hospitalized due to suspected selective intrauterine growth restriction (sIUGR) in one of the fetuses. This was a diagnostic stay in a tertiary center that is experienced in the management of twin pregnancies. The stay in the hospital made it possible to coordinate specialist consultations and perform reference ultrasound examinations. An ultrasound scan and fetal biometry were performed. Fetal well-being was assessed with an ultrasound Doppler examination of the middle cerebral artery (MCA) and umbilical artery (UA). The cerebroplacental ratio (CPR) and amniotic fluid index (AFI) were calculated (see Table 2 and Table 3). Both fetuses were in a transverse position, with details presented in Figure 3 and Figure 4. The abovementioned scan revealed a growth disturbance in one of the fetuses, and type 1 sIUGR was confirmed. A CTG recording showed deep, prolonged late deceleration in both fetuses (heart rate <80 beats/minute, lasting 7 minutes). The deceleration resolved after an intravenous injection of 25 μg of fenoterol. According to the FIGO consensus guidelines on intrapartum fetal monitoring, the risk of asphyxia was diagnosed, and the patients qualified for an emergency caesarean section (Figure 5) FIGO consensus guidelines on intrapartum fetal monitoring: Cardiotocography, 2015 [5] A female (1800 g, length 42 cm) and a male (2040 g, length 46 cm) neonate were born. The puerperium was uneventful. The patient breastfed for 4 weeks. 

## 3. Discussion

Swyer syndrome is still infrequent. The phenotype of patients with Swyer syndrome is female, but the genotype is male. The external genitalia are normal. Moreover, the vagina is normal, but the uterus is usually hypoplastic, and the gonads are dysgenetic. The most common and documented symptom of Swyer syndrome is primary amenorrhea. Treatment of Swyer syndrome requires estrogen therapy to induce puberty and long-term combined replacement therapy [8,9,10,11,12,13,14,15]. One of the main clinical problems in women with Swyer syndrome is related to reproductive health issues. Patients with Swyer syndrome rarely become pregnant, and twin pregnancies are especially rare. As the gonads of patients with Swyer syndrome are not functional, they are infertile. However, with the use of assisted reproductive techniques and proper preparation of the uterus, patients may become pregnant via egg donation and in vitro fertilization [16].

Each feature of our patient was in accordance with the respective descriptions in the literature. A well-chosen HRT is necessary for the patient to develop normal sex characteristics and to prepare the patient to seek reproduction. Before the egg donation procedure, we recommended a mini-hysteroscopy to ensure that the shape and size of the uterine cavity was correct.

As described in the available literature, patients with Swyer syndrome present with an increased risk of gonadal cancer (gonadoblastoma), which could be a precursor of dysgerminoma. The risk is estimated at about 30% [16]. Therefore, prophylactic removal of the dysgenetic gonads is recommended [8,9,10]. The timing depends on the age at diagnosis [11], but gonadal tumors may develop at any age, including during childhood. HRT therapy must be introduced after the removal of the gonads. Dural et al. [14] described the possibility of a diagnosis of Swyer syndrome in the presence of normal pubertal development and normal sex steroid levels, which is considered to be produced by gonadoblastoma [14]. The side effects of ovary removal mostly include osteoporosis and hypoestrogenism. HRT promotes the development of breasts, regulates menstrual cycles, and prevents uterine hypoplasia. 

Women with Swyer syndrome present with a hypoplastic uterus. Pregnancy might be achieved using hormone therapy and assisted reproduction techniques with egg donation. The studies conducted by Van der Hoorn et al., 2010, and Savasi et al., 2016, showed that oocyte donation (OD) pregnancies were at an increased risk of obstetric complications, especially pre-eclampsia (PE) [17,18]. 

Specific complications, such as hypertension, pre-eclampsia, and proteinuria, may be associated with pregnancies in Swyer syndrome. Taneja et al. [19], Sauer et al. [20], Kan et al. [21], Ko et al. [22], and Michala et al. [2] reported gestational hypertension and/or pre-eclampsia (see Table 4). Case reports published by Tulic et al. [23] and Cretsas et al. [24] described patients with oligohydramnios. Cases reported by Selvaraj et al. [25], Lutjen et al. [26], Cornet et al. [27], Dirnfeld et al. [28], Plante and Fritz [29], and Chen et al. [15] did not develop any complications during pregnancy. Further information on the remaining cases reported by Bardeguez et al. [30] and Bianco et al. [31] was not included. Obesity, maternal age (40 years old), and in vitro fertilization procedure constituted additional risk factors in our patient. To the best of our knowledge, we present the first case of a successful pregnancy achieved in a patient with Swyer syndrome with intrauterine growth restriction in one of the fetuses. Cesarean section was the mode of delivery in the predominant number of cases [32,33,34,35,36,37,38]. Only three patients delivered vaginally, i.e., two cases reported by Michala et al. [2] and one case by Kan et al. [21]. The vast majority of patients delivered a baby at term, between 37 and 41 weeks of pregnancy. Patients described by Sauer et al. [20], Kan et al. [21], Ko et al. [22], and Michala et al. [2], as well as our patient, delivered preterm neonates before 37 weeks of pregnancy. Most cases of Swyer syndrome are sporadic, but some familial cases have also been described [39]. Gupta et al. [39] described two sisters who presented with primary infertility and conceived following in vitro fertilization (IVF) with ovum donation. The results of pregnancies in patients with Swyer syndrome are reviewed in Table 4.

Despite the available descriptions, a model of care for pregnant patients with Swyer syndrome has not been developed yet. Furthermore, all the descriptions are rather empirical. In our opinion, one must establish patterns, thereby facilitating the management of such patients in order to avoid pregnancy complications as much as possible.

## 4. Conclusions

The treatment of Swyer syndrome requires a multidisciplinary approach. The prophylaxis of malignancy, osteoporosis, induction of puberty, fertility, and psychological support are important aspects of patients’ lives. The aim of this case report and review is to show clinicians that patients with Swyer syndrome may become pregnant without any major problems through assisted reproduction. Due to certain abnormalities in organ development, i.e., a less mature uterus, such cases may be higher-risk pregnancies. The above case report indicates the need to systematize methods of pregnancy management in patients with Swyer syndrome.

## Figures and Tables

**Figure 1 ijerph-19-05027-f001:**
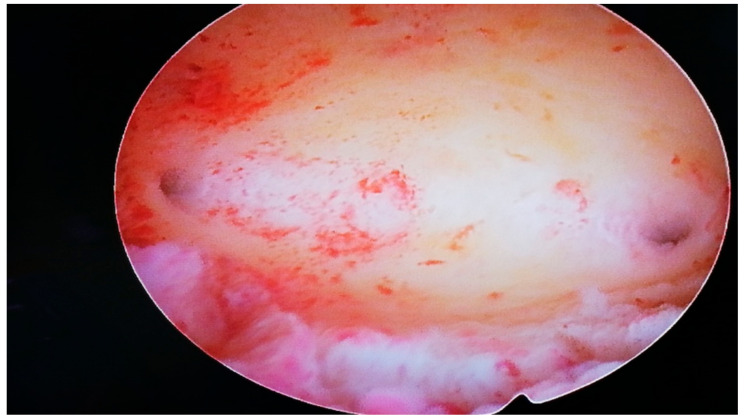
Mini-hysteroscopy view in the patient with Swyer syndrome.

**Figure 2 ijerph-19-05027-f002:**
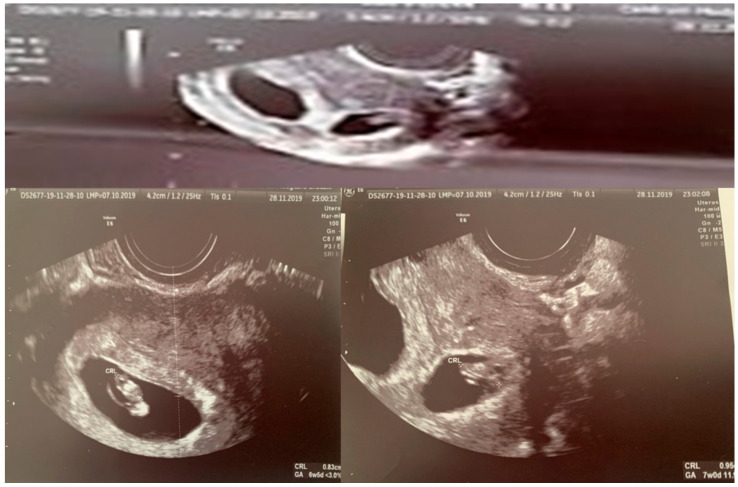
USG scan of twin pregnancy (dichorionic diamniotic—7 week of pregnancy) in patient with Swyer syndrome.

**Figure 3 ijerph-19-05027-f003:**
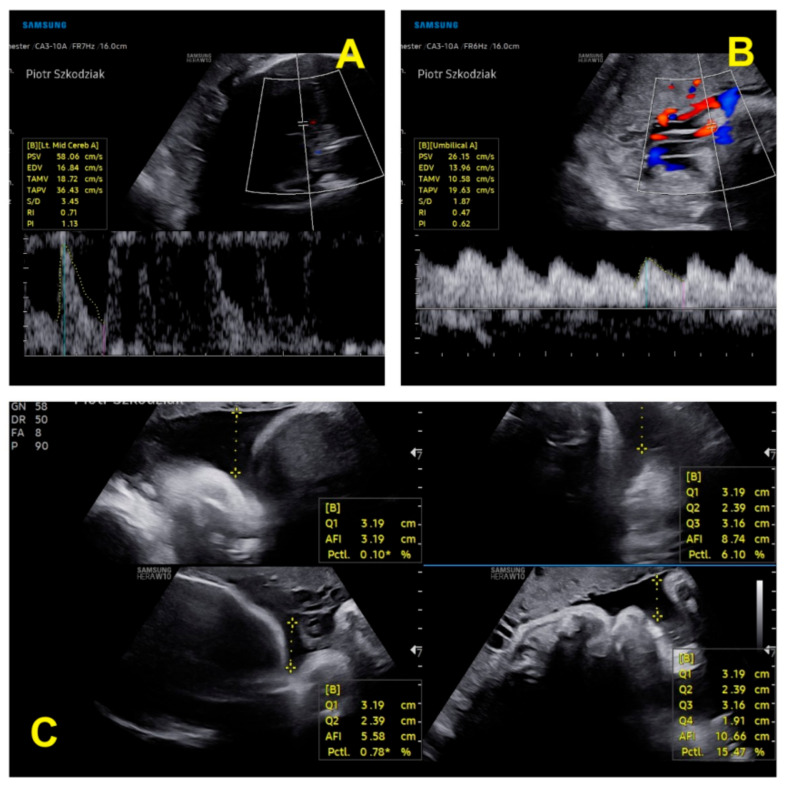
Ultrasound parameters of the first twin’s well-being: (**A**) assessment of blood flow in the MCA; (**B**) assessment of blood flow in the UA; (**C**) amniotic fluid index [6,7].

**Figure 4 ijerph-19-05027-f004:**
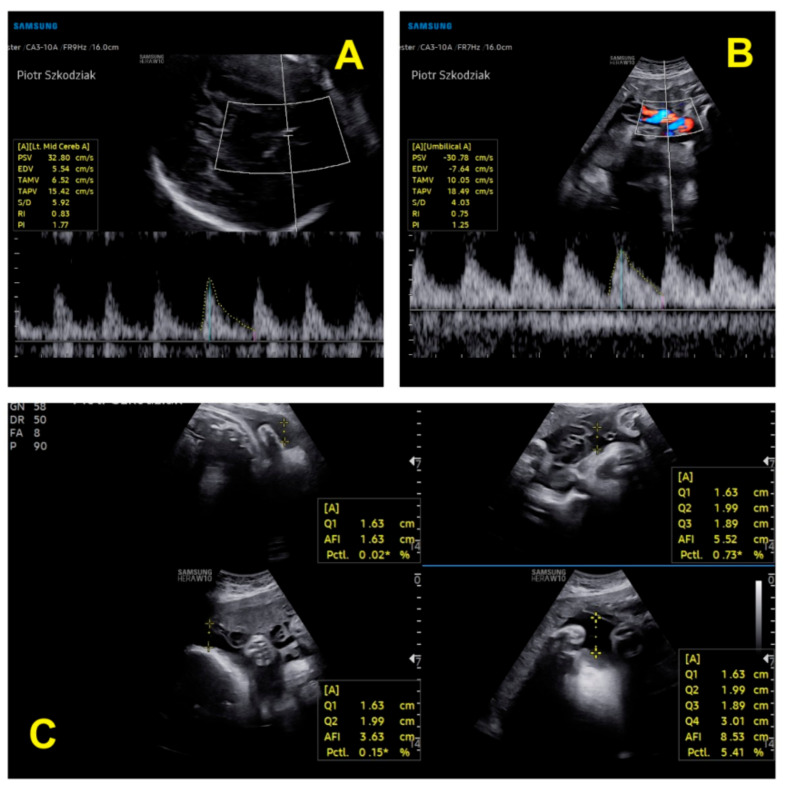
Ultrasound parameters of the second twin’s well-being: (**A**) assessment of blood flow in the MCA; (**B**) assessment of blood flow in the UA; (**C**) amniotic fluid index [6,7].

**Figure 5 ijerph-19-05027-f005:**
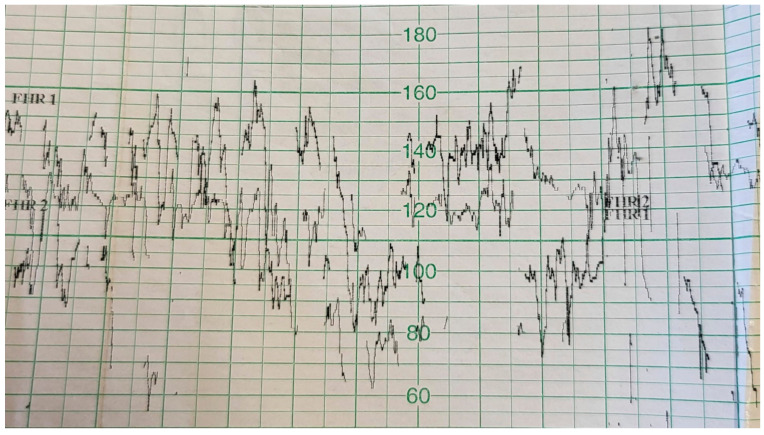
CTG recordings showed prolonged deceleration of both twins, lasting 7 minutes, and a decrease in the heart rate of both fetuses to 80 beats/minute (FIGO consensus guidelines on intrapartum fetal monitoring: Cardiotocography, 2015).

**Table 1 ijerph-19-05027-t001:** Pharmacotherapy in the presented case.

Parameter	I Trimester	II Trimester	III Trimester
Estrofem (estradiol) orally	2 mg + 2 mg + 2 mg	2 mg + 0 + 2 mg	1 mg + 0 + 1 mg
Progesterone Prolutex (progesterone) Progesterone 200 mg intravaginally	1 × 25 mg i.m./s.c. for every third day 200 mg + 200 mg + 200 mg	1 × 25 mg i.m./s.c. for every fifth day 200 mg + 0 + 200 mg	1 × 25 mg i.m./s.c. for one in week 0 + 0 + 200 mg
Encorton (prednisone)	5 mg	5 mg	2.5 mg
Aspirin (acetylsalicylic acid)	150 mg	150 mg	150 mg until its discontinuation at week 32
Other drugs	Pimafucin (Natamycin 100 mg) 1 × 1 tablet intravaginally for 6 days	Monural (fosfomycin 3 g) orally	Essentiale forte (phospholipids 300 mg − 1 tablet) 3 × 2 tablets

**Table 2 ijerph-19-05027-t002:** First trimester scan.

Parameter	Twin 1	Twin 2
**Chorionicity**	**Dichorionic Diamniotic**	
CRL(crown–rump length) (mm)	49.2–11w5d	51.30–11w6
BPD (biparietal diameter) (mm)	14.5–12w0	17.2–12w4d
FHR/min (fetal heart rate)	160	161
NT (mm) (nuchal translucency)	1.42	1.2

**Table 3 ijerph-19-05027-t003:** Ultrasound growth patterns in the second and third trimesters.

Parameter	Twin 1 in Trimester II	Twin 2 in Trimester II	Twin 1 in Trimester III	Twin 2 in Trimester III
BPD (mm) (biparietal diameter)	58.7–23w1d	57–22w5d	76.4–29w3d	75.3–29w0d
HC (mm) (head circumference)	221–24w1d	207–22w6d	275.3–29w3d	261–27w6d
FL (femur length) (mm)	40.8 mm–23w1d	37.6–22w0	54.9–29w2d	51.6–27w5
AC (abdominal circumference) (mm)	180	174	240	221.5
AFI (amniotic fluid)	normal	normal	normal	normal
FHR/min (fetal heart rate)	153	160	150	148
EFW (g) (estimated fetal weight)	570	493	1298	1057
EFW discrepancy	13.5% (assessment at 22 weeks 6 days)		18.6% (assessment at 28 weeks 0 days)	

**Table 4 ijerph-19-05027-t004:** Cases of pregnancies in patients with Swyer syndrome.

Authors	Individuals	Age at Delivery	Pregnancy Single/Multiple	Mode of Delivery	Week at Delivery	Complications
Taneja et al., 2014 [19]	1	39	single	cesarean delivery	39	pre-eclampsia
Selvaraj et al., 2002 [25]	1	27	single	cesarean delivery	not known	none
Lutjen P. 1984 [26]	1	25	single	cesarean delivery	38	none
Cornet et al., 1990 [27]	1	25	single	cesarean delivery	36-41	none
Bardeguez et al., 1990 [30]	1	unknown	triplet	cesarean delivery	Unknown	unknown
Bianco et al., 1992 [31]	1	unknown	single	not known	Unknown	unknown
Kan et al., 1997 [21]	2	30, 32 ( the same woman, two attempts)	1. single 2. twin	1. vaginal delivery 2. cesarean delivery	1. >37 2. 36 weeks	1. pre-eclampsia 2. blood pressure elevated
Sauer et al., 1989 [20]	1	28	twin	cesarean delivery	35	pregnancy-induced hypertension
Ko et al., 2007 [22]	1	35	triplet (twins alive, boy and girl)	cesarean delivery	33	pre-eclampsia proteinuria, mola hydatidosa during pregnancy
Dirnfeld et al., 2000 [28]	2	30 (the same woman, two attempts)	1. single 2. twin	cesarean delivery (both)	1. 41 2. term delivery	1. spontaneous demise of one fetus occurred at 19 weeks of pregnancy 2. none
Plante and Fritz et al., 2008 [29]	1	27	single	cesarean delivery	38	none
Tulic et al., 2011 [23]	1	30	single	cesarean delivery	39	reduced amniotic fluid
Chen et al., 2005 [15]	1	31	twin	cesarean delivery	36	none
Frydman et al., 1988 [34]	1	37	single	cesarean delivery	41	none
Michala et al., 2008 [2]	3	unknown	single	1 and 2 vaginal deliveries 3. cesarean delivery	1, 2. unknown 3. 36	1, 2. none 3. pre-eclampsia
Creatsas et al., 2011 [24]	1	35	single	cesarean delivery	38	oligohydramnios and abdominal circumference below the 5th percentile
De Santis et al., 2013 [33]	1	27	twin	cesarean delivery	35	none
Gao et al., 2011 [34]	1	32	twin	cesarean delivery	Unknown	none
Murtinger et al., 2013 [35]	1	30	twin	cesarean delivery	36	none
Kalra et al., 2016 [36]	1	27	single	cesarean delivery	34	severe pre-eclampsia
Shah et al., 2018 [37]	1	32	twin	cesarean delivery	34	hypertension
Urban et al. [38]	2	32,34	single	cesarean delivery	40, 39	no medical evidence of uterine dilatation efficacy
Gupta et al. [39]	2	25, unknown(younger)—two sisters	Single, single	Details not known	Details not known	Details not known

## Data Availability

The datasets used and/or analyzed in the current study are available from the corresponding author on reasonable request.

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
