# Peer review of "A Successful New Case of Twin Pregnancy in a Patient with Swyer Syndrome—An Up-to-Date Review on the Incidence and Outcome of Twin/Multiple Gestations in the Pure 46,XY Gonadal Dysgenesis"

_ijerph, 2022, doi:10.3390/ijerph19095027_

Round 1

Reviewer 1 Report

Line 65:  the Authors must add ....during pregnancy

Line 100  The Authors missed some words

Line 120: change with this syndrome

Line 120,121: transfer Patients with Swyer syndrome... are especially rare, to line 130 after issues

Line 124 to 127: The authors must cancel from ...Due to......to  diagnosed(8) because this concept it is considered at the next files.

Line 133  transfer to line 124 after dysgenetic

Line 143. recommended(5,6,7)

Line 144 diagnosis (8)

Line 151 Van Der Hoorn et al.,2010     Savasi et al.,2016

Lines 151 to 153 transfer to line 157 after donation

Line 153 and 154 transfer to line 165

Line 179. cancel from this is an error to line 180

Line 180: continue....In our opinion one must establish......

Line 190.    i.e., a less mature uterus

Line 192-193. I suggest to cancel....our case is well.....in the literature.

The Authors,owing the changes they must consider in the text, must provide to correct the numbers of references

Author Response

Thank You for your comment. We changed the above mentioned part of the article. All changes are in red colors according instructions of reviewer.

Reviewer 2 Report

Please provide the source of Recommendation

according to which you diagnosed fetal asphyxia in CTG. 

Is the presented Ctg a 5 minut bradycardia or

a zig-zag pattern?

Author Response

CTG recording showed deep, prolonged late deceleration in both fetuses (heart rate <80 beats/minute, lasting 7 minutes). Deceleration resolved after intravenously injection of 25 μg of fenoterol. According to the FIGO consensus guidelines on intrapartum fetal monitoring, the risk of asphyxia was diagnosed and patienrs was qualified for an emergency caesarean section.   https://doi.org/10.1016/j.ijgo.2015.06.020   https://obgyn.onlinelibrary.wiley.com/doi/10.1016/j.ijgo.2015.06.020

This manuscript is a resubmission of an earlier submission. The following is a list of the peer review reports and author responses from that submission.

Round 1

Reviewer 1 Report

1)The article,as presented, seems to make up no original contribution,in that pregnancy in patients with Swyer syndrome has been reported in literature since 1989(exactly, in 23 cases including 10 twin/multiple gestations according to the table shown in the text). So,the Authors conclusion that "the aim of this case report is tho show clinicians that patients with Swyer syndrome may get pregnant without any major problems through assisted reproduction" is a weak affirmation.

In my opinion, the Authors first, should report such a case in a more complete form and second, to review particularly the clinical aspects only of twin/multiple pregnancy in patients with this syndrome. In such a way the will be to able to answer about the question they posed, namely how twin/multiple pregnancy management should be codified in patients with Swyer syndrome(they could take in account the outcome of eleven,including their case,twin/multiple pregnancy reported in the literature). And this is important owing the concept: a)the twin/multiple pregnancy,as we know,are at risk for preeclampsia,preterm birth and selective growth restriction - the last event observed for the first time by the Authors;b) but we don't know, as  well as  potential complications are concerned, which is the actual role played by pregnancy itself, by assisted reproduction and by maternal factors(like uterus hypoplasia).

Therefore,as hypothesis, the title of article could be for instance:"A successful new case ow twin pregnancy in a patient with Swyer syndrome-un up to-date review on the incidence and outcome of twin/multiple gestations in the pure 46/xy gonadal dysgenesis". In a such way,the title could be more interesting from a clinical point of view.

2)About the text, introduction is to long and some concepts must be transferred in the Discussion part of the article.

3)More details are needed to present this case report. Among these details, it is important to report the the ultrasound scan of first trimester showing chorionicity,in that without this detail table 2 appears as incomplete;moreover,chorionicity is fundamental for the correct  twin/multiple management and to evaluate the maternal-fetal risks too.

4)Data in table 1 are not complete, namely what the Authors write in the text is not reported in the same table.However,all the tables need to be checked.

5)Concerning the figures, it is important to show also data about fetal status that represented the indication for cesarean section.

6)English language is not appropriate,so it needs to be revised.

References should be presented in a homogenous  pattern(Year,volume,pages,DOI,etc)

FINAL COMMENT

THE ARTICJE WILL BE RECONSIDERED AFTER MAJOR REVISIONS

Author Response

Dear Editor,

Please find below the response to reviewers together with detailed description of manuscript changes.

With kind regards.

Izabela Winkler

Reviewer #1:

 Comment 1

Response: We have changed the title according to Reviewer’s suggestion.

A successful new case of twin pregnancy in a patient with Swyer syndrome - an up to-date review on the incidence and outcome of twin/multiple gestations in the pure 46/xy gonadal dysgenesis"

Comment 2

 About the text, introduction is too long and some concepts must be transferred in the Discussion part of the article.

Response: We have moved a part of introduction section to discussion.

Comment 3

More details are needed to present this case report. Among these details, it is important to report the the ultrasound scan of first trimester showing chorionicity,in that without this detail table 2 appears as incomplete;moreover,chorionicity is fundamental for the correct  twin/multiple management and to evaluate the maternal-fetal risks too.

Response: We have added the necessary information in the text.

Comment  4

Data in table 1 are not complete, namely what the Authors write in the text is not reported in the same table. However,all the tables need to be checked.

Response: We have edited the Tables as suggested.

Comment 5

Concerning the figures, it is important to show also data about fetal status that represented the indication for cesarean section.

Response: We have added the necessary info in Figure. 4

Two days later, due to the high risk of fetal asphyxia (CTG recordings showed prolonged deceleration of both twins, lasting 5 minutes, and a decrease in the heart rate of both fetuses to 60 beats/minute) the patient was qualified for an emergency cesarean section (Figure 4).

Comment 6

English language is not appropriate, so it needs to be revise

Response: The manuscript has been extensively revised by specialistic MDPI service.

Reviewer 2 Report

  1. Extensive editing of English language and style required. The sentence: As the gonads of patients with Sweyer syndrome do not work properly- is unexpectable.
  2. There are no case report citations after 2018- however, there are at least 2 cases described.

Author Response

Reviewer #2:

Comment 1

Extensive editing of English language and style required.

Response: The manuscript has been extensively revised by specialistic MDPI service.

Comment 2

The sentence: As the gonads of patients with Sweyer syndrome do not work properly- is unexpectable.

Response: We changed sentences “The uterus and fallopian tubes of such patients may be smaller or normally-formed, but the gonads are not functional.

Comment 3

There are no case report citations after 2018- however, there are at least 2 cases described.

Response: We added 2 cases described after 2018.

  1. Urban, A.; Knap-Wielgus, W.; Grymowicz, M.; Smolarczyk R. Two successful pregnancies after in vitro fertilisation with oocyte donation in a patient with Swyer syndrome - a case report. Prz Menopauzalny. 2021, 20(3):158-161. doi: 10.5114/pm.2021.109361.
  2. Gupta, A.; Bajaj R.; Jindal UN. A rare case of Swyer syndrome in two sisters with successful pregnancy outcome in both. J Hum Reprod Sci 2019; 12: 267-269.

Reviewer 3 Report

Thank you for allowing me to review this interesting case. I think this case report will be interesting and informative to our readers. This case will allow readers to learn about how pregnancy can be achieved in patients with Swyer syndrome and how antenatal care should differ from regular pregnant patients. I have a few minor comments.

  1. Is it routine practice to monitor liver enzymes in patients using hormone replacement therapy?
  2. What was the prednisone 5mg po daily for?
  3. Could you describe more about the reproductive endocrinology and infertility portion of this case? e.g. How was the donor oocyte selected? What kind of protocol was used for egg retrieval, or endometrial preparation? 
  4. Page 3. betamethasone was used for fetal "lung" maturation
  5. Page 3. Instead of saying, cesarean section was performed due to risk of fetal asphyxia, could you describe in a more standardized format? e.g. What type of deceleration was it? Was it recurrent or prolonged?Was it present in one or both fetuses? 
  6. What was the indication for admission to the hospital? Is sIUGR an indication for hospitalization?

Author Response

Reviewer #3:

Comment 1

  1. Is it routine practice to monitor liver enzymes in patients using hormone replacement therapy?

Response: This is not a routine practice, however due to several risk factors of pre-eclamspia it was decided to check liver enzymes for for potential changes. We have added the necessary info in the text.

Comment 2

What was the prednisone 5mg po daily for?

Prednisone (Encorton) was used because of the immunosuppressive effect associated with oocyte donation.

Response:

Comment 3

Could you describe more about the reproductive endocrinology and infertility portion of this case? e.g. How was the donor oocyte selected? What kind of protocol was used for egg retrieval, or endometrial preparation? 

The patient received an oocyte from a Greek woman who was phenotypically similar to a patient with Sweyer syndrome. Before in vitro procedure she got Intralipid 20% in 500 ml bag in 2 hours. For endometral preparation she was taken Estrofem 3 x 2 tabl per day orally, Prolutex 1amp for 3 days, progesterone  per vaginam 3 x 2 tabl and orally progesterone 200mg 3 x 1 tabl

Response:

Comment 4

Page 3. betamethasone was used for fetal "lung" maturation

Response: The twin pregnancy described in the manuscript was treated as a pregnancy with a high risk of premature delivery when the differential weight of the fetuses was diagnosed.  Development of a sIUGR in one fetus may have resulted in the intrauterine death of one fetus.  For the above indications, prenatal steroid therapy was used to accelerate the maturation of the lungs in the fetus. We have edited this part. Thank you for this comment.

Comment 5

Page 3. Instead of saying, cesarean section was performed due to risk of fetal asphyxia, could you describe in a more standardized format? e.g. What type of deceleration was it? Was it recurrent or prolonged?Was it present in one or both fetuses? 

Response: Two days later, due to the high risk of fetal asphyxia (CTG recordings showed prolonged deceleration of both twins, lasting 5 minutes, and a decrease in the heart rate of both fetuses to 60 beats/minute) the patient was qualified for an emergency cesarean section (Figure 4).

Comment 6

What was the indication for admission to the hospital? Is sIUGR an indication for hospitalization?

Response: In this case, admission to hospital resulted from the need to extend diagnostics and issue an expert opinionsIUGR is not a standard reason for hospital admission, however in many cases it makes it a lot easier to coordinate specialistic consultations and expert ultrasound scans in hospital in Poland as most of the experts work there. When the differential weight of the fetuses was found, it was necessary to establish the cause.  As mentioned above, the pregnancy described in the manuscript was a pregnancy with a high risk of complications of preterm labor, including other complications typical of a twin pregnancy.  It was not fully known what the final impact on the development of pregnancy would be due to the donation of oocytes and the mother's abnormal karyotype.  Severe complications from the 3rd trimester of pregnancy (preeclampsia, hypertension, oligohydramnios and polyhydramnios) have been reported in the available pregnancy reviews of mothers with Swyer syndrome (manuscript table).  Development of a sIUGR in one fetus may have resulted in the intrauterine death of one fetus.  The fetuses were in terms of survival.  Hospitalization allowed us to carry out a thorough diagnosis and proper supervision of the pregnancy.  This was an indication for hospitalization.  The diagnosis states: 1. pregnancy of 31 weeks.  Di- di- twin pregnancy.  Mother is overweight.  Swyer syndrome in mother.  Status after IVF/ET from oocyte donation.  Suspected sIUGR.

Round 2

Reviewer 1 Report

Abstract: the Authors write that the aim of this case report is to show clinicians that patients with Swyer syndrome(SS) may became pregnant without any major problems through assisted  reproduction.

But this aspect already has been reported in the literature! 

They also say that the case demonstrated the need to systematize method of pregnancy management in patients with SS,but they do not refer to twin pregnancy neither consider which methods are concerned.

INTRODUCTION: what is written at page 2,file 17-20 must be printed  in file1, following the point.

There is no relationship between what is written in the article and that reported in Table 1,namely this table show pregnancy aspects instead of those concepts reported in the text.

No figures neither ultrasound images are reported as well as chorionicity is concerned.

DISCUSSION: the Authors repeat what has been written in Introduction part of the article, but they do not discuss the concept they proposed as the aim of their work,I mean how to manage pregnancy ,single one or twin,in patient with SS, considering the possible complications associated to twin gestation in healthy woman.

English language must be improved,

The Authors are invited to consider a major revision of the article , answering to the  questions they posed as the aims of their work.